# The Ectonucleotidases CD39 and CD73 and the Purinergic Receptor P2X4 Serve as Prognostic Markers in Non-Small Cell Lung Cancer

**DOI:** 10.3390/cancers17071142

**Published:** 2025-03-28

**Authors:** Konrad Kurowski, Sophie Nicole Prozmann, António Eduardo Cabrita Figueiredo, Jannis Heyer, Felix Kind, Karl-Moritz Schröder, Bernward Passlick, Martin Werner, Peter Bronsert, Severin Schmid

**Affiliations:** 1Institute for Surgical Pathology, Faculty of Medicine, Medical Center–University of Freiburg, 79106 Freiburg, Germany; 2Core Facility Histopathology and Digital Pathology Freiburg, Medical Center–University of Freiburg, 79106 Freiburg, Germany; 3Tumorbank Comprehensive Cancer Center Freiburg, Medical Center–University of Freiburg, 79106 Freiburg, Germany; 4Department of Thoracic Surgery, University Medical Center Freiburg, 79106 Freiburg, Germany; 5Department of Nuclear Medicine, University Medical Center Freiburg, 79106 Freiburg, Germany

**Keywords:** non-small cell lung cancer, purinergic signaling, CD39, CD73, P2X4, P2X7, tumor microenvironment, extracellular ATP

## Abstract

This study investigates the role of purinergic signaling in non-small cell lung cancer (NSCLC) and how it affects patient outcomes. Purinergic signaling involves molecules like ATP, its metabolites, the corresponding receptors, and certain enzymes called ectonucleotidases that have been shown to influence cancer growth and the antitumoral immune response. The researchers analyzed the expression of specific receptors and ectonucleotidases of the purinergic system (CD39, CD73, P2X4, and P2X7) in tumor cells and surrounding tissues from 139 patients with NSCLC. The results highlight the prognostic role of the ectonucleotidases and purinergic receptors in NSCLCs and show that high levels of CD39 and low levels of CD73 as well as high levels of P2X4 in the tumor are associated with a better survival outcome for the patient. Contrary to that, the purinergic receptor P2X7 is no predictor of patient prognosis in NSCLCs. These findings provide further evidence that targeting the ectonucleotidases could offer new treatment strategies to improve the prognosis of patients with NSCLC.

## 1. Introduction

Lung cancer constitutes the deadliest type of cancer worldwide with an estimated 1.8 million related deaths annually [1]. Approximately 80% of lung cancers are classified as non-small cell lung cancers (NSCLCs) based on their histological appearance. Adenocarcinoma (ADC), squamous cell carcinoma (SCC), and rare entities such as large cell carcinoma, neuroendocrine carcinoma or undifferentiated lung carcinoma all belong to this group. The prognosis of NSCLCs largely depends on the disease stadium at the time of diagnosis as well as on the molecular properties of the respective tumors. Concerning the latter, in the recent two decades, tumor immunology and the composition of the tumor microenvironment (TME) have gained increasing attention in lung cancer research since compelling evidence has shown the intricate role of those intimately linked compartments for the therapy and outcome of patients [2,3].

Purinergic signaling has emerged as a key player in this context [4,5]. Purine metabolites have many biological functions. For instance, adenosine triphosphate (ATP) acts as an energy storage molecule within the cell [6]. However, purine metabolites can also be found extracellularly. This is the case when they get released upon necrotic or apoptotic cell death or permeate the cell membrane with the help of connexin or pannexin hemichannels [7,8]. In the 1970s, the pioneering research of Geoffrey Burnstock and colleagues provided evidence that extracellular ATP (eATP) and its metabolites extracellular adenosine diphosphate (eADP), extracellular adenosine monophosphate (eAMP) as well as extracellular adenosine constitute an important pool of extracellular signaling molecules. eATP becomes hydrolyzed to eADP by the ectonucleotidase ectonucleoside triphosphate diphosphohydrolase-1 CD39. CD39 also catalyzes the hydrolysis of eADP to AMP. Of interest, the CD39-catalyzed reactions are the rate-limiting steps in extracellular purine turnover [9]. Subsequently, the conversion of eAMP to adenosine is catalyzed by the ectonucleotidase 5′-nucleotidase CD73 [10]. These metabolites can bind to different so-called purinergic receptors and induce a plethora of biological responses in the receptor-bearing cells [11]. From a functional point of view, three distinct classes of purinergic receptors can be distinguished [12]. P2X-receptors with its seven subtypes P2X1 to P2X7 are ligand-gated, non-specific cation-channels and mainly bind eATP [13]. Independent of the respective subtype, activation of P2X receptors leads to an influx of sodium and calcium into the cell and an efflux of potassium out of the cell. These chances in the membrane potential trigger different intracellular signaling pathways such as Rho-kinase pathway or Src-ERK1/2 signaling [14]. P2Y-receptors belong to the huge family of G-protein-coupled receptors and exist in eight different subtypes (P2Y1, P2Y2, P2Y4, P2Y6, and P2Y11-14). P2Y-receptors mainly bind eATP and eADP [15]. P1 receptors, represented by four subtypes, are also G-protein-coupled receptors and specifically bind extracellular adenosine [16]. Due to this complex network of low molecular weight metabolites, different ectonucleotidases and receptors, the exact type of molecular response induced by purine metabolites depends on the metabolite concentration itself as well as on the cell-type specific expression of ectonucleotidases and receptors.

In recent years, a growing body of scientific literature suggests that purinergic signaling systems are dysregulated in the TME of many malignancies, as well as in cancer cells themselves [4,5]. Although the purinergic landscape in NSCLCs has also been intensively investigated, the prognostic significance of the expression of purinergic components is not yet clearly determined. Within the TME of various cancers, eATP is mainly released by active or passive secretion from living cells or activated and dying immune cells, along with dying tumor and stromal cells [17]. Generally, it is believed that eATP is a strong activator of the antineoplastic immune response and that its degradation by ectonucleotidases contributes to immunoevasive mechanisms within the TME [18].

In NSCLCs, the ectonucleotidase CD39 was found to be expressed on tumor cells, tumor-associated endothelial cells, as well as on cells of the immune systems including B-cells, regulatory and activated CD4^+^ or CD8^+^ T-cells, and innate immune cells [19,20]. Histopathologic examinations of surgical specimens from NSCLC patients have shown an increase in CD39 expression in the TME in comparison with the healthy lung tissue [21]. The authors of the latter study found an intriguing co-expression of the immune checkpoint protein programmed cell death protein 1 (PD1) with CD39 in patients with tumor relapse, suggesting a role of CD39 in tumor-related immunoevasive mechanisms. In this context, another study could show that a high fraction of CD39^+^ CD8^+^ (doubly positive) T-cells within the TME of NSCLCs is a predictive biomarker for patients being responsive to immune checkpoint inhibitor treatment, underscoring its role in the antineoplastic immune response [22]. However, Shao et al. showed an increase in PFS and overall survival (OS) in ADC patients with a high CD39 expression [23]. Based on these contradictory results, the exact pathophysiological and prognostic role of CD39 in NSCLC patients is still incompletely understood and of great therapeutic interest.

CD73 expression in NSCLCs was found on cancer cells themselves as well as on tumor-infiltrating lymphocytes and cancer-associated fibroblasts within the TME. Like CD39, CD73 contributes to the immunosuppressive milieu within the TME. One proposed mechanism is the CD73-dependent upregulation of PD-ligand 1 (PD-L1) in cancer cells [24]. So far, clinical studies are ambiguous concerning the prognostic significance of CD73 expression in NSCLCs, although some suggested that a high CD73 abundance might be an independent indicator of an unfavorable prognosis in NSCLCs patients [25,26,27].

Expression analyses proved a differential regulation of P2X-family purinergic receptors of cell lines established from NSCLC patients [28]. Most studies conclude that P2X-receptores are highly overexpressed in lung cancer cell lines and that a constitutive ATP release from the cells together with a high receptor density of, e.g., P2X7 form an autocrine signaling loop that promotes the proliferation of the malignant cells [28]. Especially for subtypes P2X4 and P2X7, a strong association of an increased RNA expression with distant metastasis was reported, being indicative of a poor prognosis [29]. Apart from that, T-cells express both P2X4 and P2X7 receptors, whereas macrophages only possess P2X7 receptors, shaping a highly specialized immunological TME [30,31]. The prognostic and therapeutic significance of the latter finding, however, remains unclear.

Monoclonal antibodies directed against CD39 (SRF617) and CD73 (Oleclumab) are currently evaluated for clinical use in the NSCLC context [32,33]. Above, experimental low molecular weight modulators of P2X receptors become increasingly available, making future clinical use possible [34]. It is thus highly desirable to better understand the pathophysiological role and prognostic significance of both ectonucleotidases and the P2X-family receptors in this patient group. Above all, it is so far unknown, if there is a significant and clinically relevant difference between the abundance of the above-mentioned proteins between the tumor stroma and the cancer cells themselves or between SCCs and ADCs. To answer these questions and to further elucidate the correlation of purinergic signaling molecule expression and disease trajectories, we analyzed the CD39, CD73, P2X4, and P2X7 expression levels in NSCLC tumor samples of 139 patients. Contrary to existing studies, we analyzed the expression data separately for tumor cells and the surrounding TME as well as for SCCs and ADCs and correlated them with clinicopathological and survival data. Our study therefore adds important findings to the existing body of evidence concerning the important role of the purinergic signaling system in NSCLCs.

## 2. Materials and Methods

### 2.1. Human Tissue Samples

The study was approved by the ethics committee of the Faculty of Medicine, Albert-Ludwigs-Universität Freiburg (REF 20-289). All tumor specimens were resected by the Department of Thoracic Surgery at the Medical Center Freiburg between 2014 and 2017, with written informed consent obtained from all patients prior to inclusion.

A total of 139 samples from patients with NSCLC were included in the study. All tissue specimens underwent processing for diagnostics according to standardized routine protocols, including sectioning, 24 h formalin fixation, paraffin embedding, cutting, hematoxylin-eosin (H&E) staining, and evaluation by board-certified pathologists at the Institute for Surgical Pathology, Medical Center Freiburg, Germany (ISP). The WHO entity, ADC vs. SCC, was immunohistochemically proven using immunohistochemical staining for p40 and TTF1.

For study inclusion, formalin fixed and paraffin embedded tissue specimens, and the corresponding slides were retrieved from the ISP archive and re-evaluated according to the latest UICC and WHO classifications [35]. Subsequently, all tissue specimens and the corresponding clinicopathological data were pseudonymized. Clinicopathological data comprised TNM-classification, WHO-classification, R-Status, sex, age at the time of first NSCLC-diagnosis, pleural infiltration, tumor size, neoadjuvant therapy, smoking history, the maximum standardized uptake value (SUVmax), disease relapse, and the patients’ progression-free survival (PFS).

### 2.2. Tissue Microarray Construction

Tissue microarrays (TMAs) were constructed as previously described [36]. In short, digitized and pseudonymized H&E slides (PANNORAMIC^®^ 1000-Scanner, 3DHISTECH Ltd., Budapest, Hungary) were uploaded to a local server (Casecenter, Sysmex, Service Unit v. 2.0903111521). For each patient, three circular tumor regions of interest (ROIs) with a diameter of 1.0 mm each, were annotated by experienced pathologists using the Caseviewer software (Sysmex, v. 2.3). TMAs were then constructed using the TMA Grand Master (3DHISTECH Ltd., Budapest, Hungary). For TMA design, an asymmetric array with 11 columns and 17 rows was selected for distinct orientation. To prevent batch effects, the acceptor TMA arrays were designed and transferred randomly. Additionally, each TMA received on-slide control cores (three placenta and tonsil specimens, respectively, per TMA acceptor block). After completing the transfer process, the acceptor block was placed on a glass slide, heated to 37 °C for 60 min, and then immediately cooled on crushed ice for 10 min. This procedure was repeated five more times.

### 2.3. Immunohistochemistry

Immunohistochemical staining was conducted using commercially available primary antibodies: P2X7 (P2X7, Novusbio, Centennial, CO, USA, polyclonal Rabbit Clone, #NBP2-19654), P2X4 (P2X4, Invitrogen, Carlsbad, CA, USA, polyclonal Rabbit Clone, #PA5-83466), NT5E/CD73 (NT5E/CD73 (D7F9A), CellSignaling, Danvers, MA, USA, monoclonal Rabbit Clone, #13160S), Ectonucleoside triphosphate diphosphohydrolase 1 recombinant protein epitope signature tag/CD39 (PrEST)/CD39 (CD39, Sigma, St. Louis, MO, USA, polyclonal Rabbit Clone, #HPA014067). Tissue sections of 2 µm thickness were cut (Leica RM2255 rotary microtome, Leica Biosystems, Nussloch, Germany) and deparaffinized. The staining protocol included the following steps: demasking of epitopes with demasking solution (pH 6.1, 30 min for P2X7 and CD73; pH 9, 20 min for CD39; sico5, 5 min for P2X4), incubation with blocking reagent (EnVision FLEX Peroxidase Blocking Reagent, DAKO, Santa Clara, CA, USA) for ten minutes, staining with the aforementioned primary antibodies at a concentration of 1.8 µg/mL (P2X7), 2.0 µg/mL (P2X4), 1:200 (CD73, concentration not reported by manufacturer) or 2.0 µg/mL (CD39), respectively, with rabbit linker (EnVision Flex + Rabbit Linker, DAKO) for 15 min, with horseradish peroxidase (EnVision FLEX/HRP, DAKO) for 20 min, followed by Envision FLEX diaminobenzidine (DAB) + Substrate Buffer (1:51) for 10 min, and finally counterstaining with hematoxylin.

### 2.4. Tissue Microarray Analysis

After staining, all TMA slides were digitized (Panoramic Scan 150, Sysmex) and imported into QuPath (v. 0.3.2) [37] for the image analysis, as previously described [36]. In short, the QuPath TMA dearrayer was applied to identify TMA cores, the TMA grid was adjusted by discarding missing cores or cores with prominent artifacts. Each core was annotated with the respective patient-identifiers by importing the TMA map. QuPath performed an automated color deconvolution for proper stain separation.

Distinct cells and their cytoplasmic immunohistochemical staining were detected with the “Positive cell detection” command. Staining-intensity was identified as 0 = negative, 1+ = weak, 2+ = moderate, and 3+ = strong. The thresholds for the scores were set by two pathologists with regard to overall staining intensity. To comprise relative staining frequency and intensity, an H-score [(percentage of cells stained 1+ ×1) + (percentage of cells stained 2+ ×2) + (percentage of cells stained 3+ ×3); values ranging from 0 to 300] was calculated.

In order to separate tumor and stroma cells, a random tree cell classifier was trained by manually annotating a subset of tumor and stroma cells across various TMA slides. The classifier was applied for all TMA cores. To obviate misinterpretations, all automatically evaluated classifications were reviewed by two pathologists. In case of misinterpretation, the relevant core was manually corrected, and the random tree cell classifier was re-trained. Finally, H-Scores from all included tumor and stroma cells derived from each TMA core were exported as a text file.

### 2.5. Statistical Analysis

Statistical analyses were conducted using R (v. 4.2.2, The R Foundation for Statistical Computing, Vienna, Austria) and R-Studio (desktop v. 2022.07.2 + 576, RStudio, PBC, Boston, MA, USA). A custom coded script, as previously described, was applied [36]. In short, for progression-free survival estimation, the survival (v. 3.2-13), gtsummary (v. 2.0.1) and cutpointr (v. 1.1.2) R-packages were used. Patients were dichotomized into high- and low-expression groups by a cutoff of the H-score. Optimal cutoffs were determined using the cutpointr R package with the maximize_boot_metric method. This approach maximizes the sum_sens_spec metric, balancing sensitivity and specificity, and uses bootstrapping to account for variability in the data. Various thresholds of the biomarker are assessed to find the cutoff that optimizes a selected metric relative to the survival outcome, allowing for the classification of patients into high-risk and low-risk groups based on their biomarker levels. The survival curves were estimated by the Kaplan–Meier method with the log-rank test for estimating difference in co-variable depending survival analyses for antibody-expression in tumor and surrounding stroma cells, age, sex, pT-, and pN-status, lymph (L) and blood vessel invasion (V), perineural sheath invasion (Pn), histological subtype, tumor grade (G), residual disease (R), pleural infiltration, tumor size, neoadjuvant therapy, pack-years (PY), the maximum standardized uptake value (SUVmax), and disease relapse.

Multivariable survival analyses were performed using Cox Proportional Hazards Models. The proportional hazards assumption was assessed for the biomarker expression and clinicopathological covariables using the scaled Schoenfeld residuals test. Covariables that violated the assumption were stratified in the Cox model. The multivariable survival analyses were performed for the entire study group as well as separately for the SCC and ADC subgroups. Given a hypothesis-driven approach, a limited set of clinically relevant covariables, and primary endpoint analysis, no *p*-value adjustment for multiple comparisons was performed.

The value distribution was determined by rejection of Null-hypothesis with the Shapiro–Wilk-Test for tumor- and surrounding stroma-cell antibody-expression. *p*-values < 0.05 were considered statistically significant.

## 3. Results

### 3.1. Patient Characteristics

This study included 139 patients (96 male (69%); 43 female (31%)) with a mean age of 67 years at the time of surgical resection (range 35–90). Ten patients received neoadjuvant therapy (7.3%). The cohort also reported an average of 45 pack-years of smoking (range: 2–100). The mean SUVmax was 20, with a median of 12 and a range of 2–99 (Appendix A).

### 3.2. Patient Survival

Survival data revealed that at the time of the last follow-up, 49 patients (35%) were deceased, and 73 patients (53%) were deceased or experienced disease progression by the last follow-up. The remaining 66 patients (47%) were censored. The mean PFS was 44 months with a range of 2–94 months and a median of 51 months. Metachronus metastases were detected in 15 patients (11%) (Appendix A).

### 3.3. Histopathologic Parameters

In accordance with the WHO-classification (5th version) [35], the tumors were classified into ADC (75 cases, 54%) and SCC (64 cases, 46%). In accordance with the UICC-classification (8th version) [38], 33 patients (24%) were classified as pT1, 66 patients (47%) as pT2, 27 patients (19%) as pT3, and 13 patients (9%) were classified as pT4. The median tumor size was 4.2 cm with a range from 0.7 cm–10.5 cm. The carcinomas were graduated as G2 in 66 cases (47%) and G3 in 73 cases (53%). Local lymph node dissection was conducted in 138 cases (99%) with evidence of lymph node metastasis in 63 cases (pN0: 75 (54%); pN1: 34 cases (25%); pN2: 29 cases (29%)). In 53 cases lymph vessel invasion was demonstrated (L1: 38%), whereas blood vessel invasion was detected in 18 cases (V1: 13%) and perineural sheath invasion in 11 cases (Pn1: 8%). A histological complete resection (R0) was achieved in 126 patients (91%), an incomplete resection with residual disease in ten patients (R1; 7%); in three cases the complete resection could not be proven histologically (Rx, 2%). Pleural infiltration was observed in 52 patients (38%) (Appendix A).

### 3.4. Analysis of the Estimated Progressive-Free Survival Using Kaplan–Meier Method and Logrank-Test

The PFS was significantly lower with a progressive pT- and pN-classification, male sex, lymph, and blood vessel invasion as well as for patients with incomplete surgical resection. Other clinicopathological parameters (age, histological subtype, neoadjuvant therapy, pack years, Pn status, and tumor grade) revealed no significant correlations with PFS (Table 1).

### 3.5. Expression of Ectonucleotidases and Purinergic Receptors in NSCLCs

For CD39, the mean expression in tumor cells was 20% (standard deviation [SD] = 28%), with a median of 6% (range: 0–100%). The CD39 H-Score in tumor cells had a mean of 25 (SD = 43), a median of 7, and a range of 0 to 246. In stromal cells, the mean expression of CD39 was 75% (SD = 21%), with a median of 81% and a range of 13% to 100%. The mean stromal CD39 H-Score was 124 (SD = 50), with a median of 121 and a range from 14 to 250 (Figure 1, Appendix A).

Tumor cells exhibited a mean CD73 expression of 23% with a SD of 30%, a median of 9%, and a range from 0 to 96%. The corresponding H-Score showed a mean of 39 with a SD of 61, a median of 10, and a range from 0 to 228. In stromal cells, CD73’s expression had a mean of 28% with a SD of 25%, a median of 18%, and a range from 0% to 93%. The H-Score for stromal cells showed a mean of 39 with a SD of 42, a median of 22, and a range from 0 to 178 (Figure 2, Appendix A).

The P2X7 expression in tumor and stroma cells was evaluated in 138 tumor samples, since one case was excluded due to section-related artifacts. In tumor cells, the mean expression was 97.7%, with a SD of 4.7%, a median of 100%, and a range from 76.3 to 100%. The mean H-Score was 177 with a SD of 50, a median of 179, and a range from 82 to 283. In the stroma, P2X7’s mean expression was 84% with a SD of 19%, a median of 95%, and a range from 33 to 100%, while its H-Score showed a mean of 90 with a SD of 21, a median of 97, and a range from 36 to 125 (Figure 3, Appendix A).

The purinergic receptor P2X4 was evaluated in 136 tumor specimens since three cases had to be excluded due to section-related artifacts. P2X4 in tumor cells showed a mean expression of 32% with a SD of 33%, a median of 16%, and a range from 0 to 98%. The H-Score had a mean of 48 with a SD of 54, a median of 19, and a range from 0 to 223. In stromal cells, the mean expression of P2X4 was 17% with a SD of 20%, a median of 6%, and a range from 0 to 80%. Its H-Score had a mean of 21 with a SD of 25, a median of 7, and a range from 0 to 96 (Figure 4, Appendix A).

The H-scores of all protein expressions except for CD39 in stromal cells exhibited a skewed distribution. This was defined by rejection of the Null-hypothesis of the Shapiro–Wilk-Test (Table 2). Patients were dichotomized into high- and low-expression groups by a cutoff of the H-score (Table 2), determined as described above.

### 3.6. Logrank-Test of Progression-Free Survival and CD39 Expression

Considering the complete NSCLC cohort (ADC and SCC), the Logrank-test identified a significantly prolonged PFS in patients with high CD39 expression in tumor cells (*p* = 0.0058) and surrounding stroma cells (*p* = 0.0067). In the SCC subgroup, CD39 expression in tumor cells and surrounding stroma cells did not show a significantly prolonged PFS (*p* = 0.1 and *p* = 0.9, respectively). In patients with ADC, high CD39 expression in tumor cells as well as in surrounding stroma cells correlated with a prolonged PFS (*p* = 0.01 and *p* = 0.023, respectively, Figure 5).

### 3.7. Cox-Regression of Progression-Free Survival and CD39 Expression

Considering the complete NSCLC cohort (ADC and SCC), the univariable analysis revealed that a low CD39 expression in the tumor and its surrounding stroma cells was significantly associated with a worse prognosis (hazard ratio [HR]: 1.89; 95% confidence interval [CI]: [1.19, 2.99], *p* = 0.007 and HR: 2.20; 95% CI: [1.23, 3.94], *p* = 0.004, respectively). Other covariables with a significant impact were sex, pT- and pN-status lymph- and blood vessel-invasion, R-status, pleural infiltration, lymph node metastasis and tumor size (Appendix A).

In the multivariable analysis, only the CD39 expression in tumor surrounding stroma cells showed a significant correlation with the risk of death (HR: 2.49; 95% CI: [1.30, 4.76], *p* = 0.004). CD39 expression in tumor cells showed no significant correlation in the multivariable analysis. Covariables with a significant impact on PFS were sex and blood vessel-invasion (Appendix A).

In the SCC subgroup, CD39 expression in tumor cells showed no significant impact in the uni- and multivariable Cox-regression (Appendix A).

In patients with ADC, low CD39 expression in tumor as well as in surrounding stroma cells correlated with a significantly higher risk of progression (HR: 2.23; 95% CI: [1.19, 4.16], *p* = 0.014 and HR: 3.58; 95% CI: [1.1, 11.6], *p* = 0.011, respectively) in the univariable analysis. Other covariables impacting significantly the risk of progression were sex, pT- and pN-classification, lymphatic-vessel invasion, residual disease and metastatic lymph nodes. The multivariable Cox-regression did not show a significant impact of CD39 expression on the risk of progression (Appendix A).

### 3.8. Logrank-Test of Progression-Free Survival and CD73 Expression

The Logrank-test did neither identify a significant impact of CD73 expression on PFS in the whole cohort nor in the SCC or the ADC subgroup (Figure 6).

### 3.9. Cox-Regression of Progression-Free Survival and CD73 Expression

The univariable analysis of the complete NSCLC cohort did not show a significant impact of CD73 expression on risk of progression. In the multivariable model a low CD73 expression in tumor cells was significantly correlated with a reduced risk of progression (HR: 0.47; 95% CI: [0.28, 0.8], *p* = 0.004), whereas a low CD73 expression in surrounding stroma cells showed an increased risk of progression (HR: 4.26; 95% CI: [1.43, 12.7], *p* = 0.003). Other variables with a significant impact were sex, pT- and pN-classification (Appendix A).

In the SCC subgroup, the CD73 expression did not show a significant correlation with the risk of progression. In the multivariable analysis, a low CD73 expression in tumor cells showed a significantly reduced risk of progression (HR: 0.37; 95% CI: [0.15, 0.93], *p* = 0.035). No other variables showed a significant correlation in the multivariable analysis (Appendix A).

In the ADC subgroup, CD73 expression did not show a significant impact in the univariable Cox-regression. In the multivariable analysis, low CD73 expression in tumor cells showed a significantly reduced risk of progression (HR: 0.40; 95% CI: [0.19, 0.85], *p* = 0.014) (Appendix A).

### 3.10. Logrank-Test of Progression-Free Survival and P2X7 Expression

The P2X7 expression in both tumor and stroma cells did not show a significant impact on PFS in the Logrank-test. This result was independent of the analyzed group, i.e., the whole NSCLC cohort or the SCC or ADC subgroups (Figure 7).

### 3.11. Cox-Regression of Progression-Free Survival and P2X7 Expression

The univariable and multivariable model did not show a significant impact of P2X7 expression on the hazard ratio of progression (Appendix A).

### 3.12. Logrank-Test of Progression-Free Survival and P2X4 Expression

The Logrank-test did not show a significant impact of P2X4 expression on patients’ PFS (Figure 8).

### 3.13. Cox-Regression of Progression-Free Survival and P2X4 Expression

The expression of P2X4 did not show a significant effect on the risk of progression in the univariable model. Multivariable Cox-regression showed a significantly increased risk of progression in tumors with a low P2X4 expression in the tumor cells (HR: 1.95; 95% CI: [1.04, 3.65], *p* = 0.039) and a positive prognostic impact in tumors with low expression in surrounding stroma cells (HR: 0.39; 95% CI: [0.20, 0.75], *p* = 0.004). Furthermore, sex, pN-classification and blood vessel invasion showed a significant impact in the multivariable analysis (Appendix A).

In the SCC subgroup, the P2X4 expression did not show a significant correlation with the risk of progression in the univariable analysis. In the multivariable analysis, a low P2X4 expression in tumor cells showed a significantly increased risk of progression (HR: 3.39; 95% CI: [1.39, 8.26], *p* = 0.009). Other covariables showing a significant correlation in the multivariable analysis were pT- and pN-classification (Appendix A).

In the ADC subgroup, P2X4 expression did not show a significant impact in the univariable Cox-regression model. In the multivariable analysis, low P2X4 expression in tumor cells showed a significantly increased risk of progression (HR: 2.92; 95% CI: [1.23, 6.89], *p* = 0.018, Appendix A).

## 4. Discussion

The purinergic system plays a pivotal role in the tumor-associated signaling in many cancers, including tumors of the breast, colon, prostate, brain, and lung [4,5]. The different purine metabolites are important parts of the extracellular metabolome of the TME [39]. Firstly, purines have auto- and paracrine functions and are involved in positive feedback loops that include cellular ATP release and eATP-dependent cell proliferation, cell migration and induction of epithelial-to-mesenchymal transition (EMT) mediated by purinergic receptor signaling. Secondly, high levels of eATP activate purinergic receptors, such as P2X4 and P2X7, on tumor infiltrating immune cells, initiating antineoplastic immune responses [30,31]. Thirdly, however, eATP can also be extracellularly converted to eADP, eAMP, and extracellular adenosine by the ectonucleotidases CD37 and CD73. Adenosine is a strong inhibitor of the anti-tumor immune response and inhibits T effector cells, thereby contributing to cancer immunoevasion [40]. Thus, purines in the TME fulfill different roles in cancer signaling, which highly depend on the exact cellular and molecular context and can involve both tumor promoting and anti-tumor effects.

Recent research efforts have also established a central role of purinergic signaling in NSCLCs. For example, a previous study by our group that mainly analyzed immune cells in the TME showed that patients with NSCLCs have higher concentrations of CD39, P2X4, and P2X7 in bronchoalveolar lavage (BAL) fluid compared to chronic obstructive pulmonary disease (COPD) patients, suggesting a role of purinergic signaling in NSCLC [29]. However, results are limited and conflicting since many previous studies have not analyzed the expression of proteins from the purinergic system in the tumor cells and the stroma cells separately and did also not consequently distinguish between ADCs and SCCs. Therefore, our new study is an important contribution to the dynamic field of purinergic signaling research in the context of NSCLCs.

We analyzed a representative dataset from a total of 139 NSCLC patients at a University Medical Center in Germany. NSCLC tissue was stained immunohistochemically to analyze the expression of P2X4, P2X7, CD39, and CD73 within tumor cells and its surrounding stroma. This study aimed to demonstrate the impact of the purinergic system on the prognosis of NSCLC and its correlation with clinicopathological parameters. Following the worldwide trend, ADCs were more abundant than SCCs in our study population [41]. Since the samples were derived from surgically treated patients, pT1 and pT2 stages were overrepresented in comparison with the usual UICC stage at diagnosis [42], which is most likely stage III or IV. This is a potential shortcoming regarding the generalizability of our data to all NSCLC patients. As expected, V1-, L1-, or R1/2-status as well as higher pT- and pN-classification were associated with a reduced PFS. In accordance with the international literature, male study participants generally had a worse prognosis when compared to female study participants [43]. Although conducted as a single-center retrospective trial, the overall descriptive parameters of our study population indicate a high external validity.

Immunohistochemical analysis of CD39 expression revealed a relatively low fraction of positively stained tumor cells (mean value 20%) and a higher fraction of positively stained cells in the surrounding stroma (mean value 75%). Of interest, a high CD39 expression in both tumor and stroma cells were significantly associated with a prolonged PFS. This was also true for the analysis of the ADC subgroup but not for the SCC patients. The details of the results differed slightly when separately applying multivariable and univariable analysis, respectively (see Section 3.7 and Appendix A for details). Generally, the research into CD39 and its association with cancer immunology has strongly increased within the last decade, but the findings regarding CD39 are ambivalent. Previous studies showed that overexpression of CD39 is a possible escape mechanism of tumor cells to reduce the concentration of eATP, which normally induces anti-tumor activity. Therefore, an overexpression of CD39 is associated with a poor outcome and tumor progression in various solid tumors; however, not including lung cancer [44,45,46]. In NSCLCs, tumor infiltrating CD8^+^ T-cells or CD4^+^ that simultaneously express CD39 are characterized by a strong anti-tumor reactivity and have a high probability to react with neo-antigens of the tumor [47]. This finding is in line with the study results of Shao et al. As in our study cohort, they could show in a bioinformatics approach that in lung ADCs, an increased CD39 expression is associated with a better PFS and OS. The authors hypothesized that CD39 somehow helps to overcome immunoevasive mechanisms in immune cells in the TME of ADCs [23]. In line with our findings, flow cytometry studies of untreated NSCLCs by Koppensteiner et al. could show that CD39 is mainly expressed in natural killer cells, tumor-associated fibroblasts, and T-cells, i.e., cells of the TME, but less so in the tumor cells themselves. Especially the CD8^+^ CD39^+^ CD103^+^ T-cell population within the tumor nests, which exhibits an exhausted phenotype but is highly cytotoxic, was associated with a better recurrence-free survival in their study cohort. The authors suggest that CD103 expression enables these T-cells to directly invade the tumor nests. Contrary to that, CD4^+^ CD39^+^ CD103^−^ T-cells residing within the tumor stroma exhibited a highly immunosuppressive phenotype [48]. The latter findings might explain why the results on the role of CD39 in NSCLC are somewhat ambiguous and strongly depend on the exact cell population studied. Our data underscore the importance of distinguishing between ADCs and SCCs as well as to differentiate between CD39 expression in the tumor-associated stroma and the malignant cells themselves. We conclude that analysis of T-cell subpopulations within the TME could help to stratify patients according to their prognosis and to find out which patients could benefit from a treatment with monoclonal antibodies targeting CD39 (e.g., if the CD4^+^ CD39^+^ CD103^−^ dominates over the CD8^+^ CD39^+^ CD103^+^ phenotype). The anti-CD39 antibodies TTX-030 and SRF617 have been shown to lead to higher levels of eATP and lower levels of adenosine. Warren and colleagues were able to show the stabilization of high ATP-levels and low levels of adenosine in mice using SRF617, which additionally led to a lower expression of CD39 on immune cells. They demonstrated that mice treated with SRF617 had an increased infiltration of CD8^+^ and CD4^+^ T-cells in orthotopic tumors and an increase in cytokines that can attract macrophages [49]. Additionally, Perrot and colleagues showed that the use of an anti-CD39 antibody can enhance the antineoplastic activity of ATP-inducing chemotherapy agents like oxaliplatin [50]. Nonetheless, the exact molecular mechanisms by which CD39 modulates the immune system in such an intricate way remain to be elucidated.

Concerning CD73, 23% of the tumor cells (mean value) and 28% of the stroma cells (mean value) exhibited positive staining. In the Logrank-test, no statistically significant correlation between CD73 and the PFS could be detected. However, upon multivariable Cox regression, a low CD73 expression in the tumor and a high CD73 expression in the stroma correlated with a favorable patient outcome in the complete NSCLC cohort. In the SCC subgroup, the multivariable analysis revealed the same statistically significant trends. In the ADC group, only a low CD73 expression in the tumor cells was associated with a reduced risk of progression. Many findings from the literature support our study results concerning CD73. Zhu and colleagues found that CD73 promotes NSCLC metastasis by different mechanisms in ADCs and SCCs. As a membrane-bound enzyme, CD73 can be enzymatically cleaved and released into the TME. In SCCs, soluble CD73 binds to the receptor tyrosine kinase Axl and stimulates EMT via SMAD-signaling. In ADCs, the enzyme function of CD73 seems to be more important than its soluble form. In converting eAMP to adenosine, a high abundance of CD73 stimulates SMAD-signaling and EMT via P1 receptors, leading to a poor prognosis [26]. Adenosine signaling in NSCLCs has been shown to efficiently impair the cytotoxic functions of T-cells [51]. Experimental evidence published by Han et al. shows an increased CD73 expression in NSCLC patients with epidermal growth factor receptor (EGFR) mutations or anaplastic lymphoma kinase (ALK)-rearrangements. They propose that CD73 overexpression in those cancers is driven by an ERK1/2-Jun pathway, which consequently contributes to an immunoevasive TME [27]. In line with our data, a large study by Inoue and colleagues also reported the prognostic significance of CD73 expression. They investigated a cohort of 642 resected NSCLCs via immunohistochemistry and included an expression profile analysis of 133 of these tumors. A higher CD73 expression was noted in EGFR mutated tumors and CD73 was an independent risk factor for a poor prognosis and a shorter recurrence-free survival. Of note, females, those who had never smoked, and ADC patients were more likely to exhibit a high CD73 expression [25]. Supporting these findings, there have been various phase I and II studies investigating the use of monoclonal antibodies targeting ectonucleotidases in NSCLCs. The therapeutic antibody Oleclumab inhibits CD73 and leads to lower concentrations of adenosine and higher concentrations of upstream purine metabolites. In the COAST-study, Herbst and colleagues showed the beneficial effect of a combined therapy of Oleclumab with the checkpoint inhibitor Durvalumab in patients with unresectable stage III NSCLC. Patients receiving this combined therapy demonstrated higher objective response rates and a prolonged PFS compared to monotherapy with Durvalumab [32]. Similar findings were observed with the use of Durvalumab combined with Oleclumab as a neoadjuvant treatment for patients with resectable NSCLCs. In the NeoCOAST study, combination therapy led to greater pathologic response rates than Durvalumab alone [52]. Combination therapies of Oleclumab with EGFR inhibitors, however, do not yet show such promising results [53]. Of note, multiple studies showed that the combined inhibition of CD39 and CD73 increased the penetration of T-cells into tumors [54]. Therefore, a combined blockade of both ectonucleotidases could increase the antineoplastic immune response and eventually lead to prolonged survival [50]. The clinical significance of our finding that a high CD73 expression in the tumor stroma is associated with a worse prognosis needs to be further studied, since this aspect has not yet been addressed by other researchers.

Data regarding the purinergic receptor P2X4 in NSCLC are very limited. Therefore, it was highly interesting for us to see if there is any correlation between its expression and clinical outcome parameters. 32% of tumor cells (mean value) and 17% of stroma cells (mean value) showed a P2X4 staining when analyzed immunohistochemically. The Logrank-test identified no statistically significant association between P2X4 expression and PFS in our cohort. In the Cox-regression analysis, a low P2X4 expression in the stroma was associated with a better prognosis, whereas a low P2X4 expression in the tumor cells led to an increased HR. In the SCC subgroup, a low P2X4 expression in the stroma was also associated with a better prognosis. Our own previous study showed that high concentrations of P2X4 in the BAL, which mainly represents the TME, are prevalent in patients with distant metastases [29]. Our new findings therefore support the data from the BAL study. Since P2X4 is involved in the early activation of CD8^+^ cells, which are important for anti-tumor immunity, our findings appear counterintuitive. Therefore, we hypothesize that other, so far unknown, mechanisms apart from T-cell activation are important for the explanation of our results in NSCLC. It is known from other tumors like breast cancer that P2X4 can exert its pro-malignant role via regulation of EMT and autophagy [55]. Mouse models of aggressive prostate cancer with P2X4 overexpression showed promising results when targeting the protein with specific inhibitors [56]. In conclusion, these data offer new perspectives concerning P2X4’s role in NSCLCs and should entail further research in this direction.

The mean expression of P2X7 was 97.7% in tumor cells and 84% in stroma cells in our study, representing the highest percentages of positively stained cells of all four proteins tested. Neither in the Logrank-test nor in the Cox regression models did we find any statistically significant association with the patient outcome parameters. In the literature, low levels of purinergic receptors like P2X7 were shown to enable tumor cells to survive despite high concentrations of eATP [28]. Mechanistically, the authors discuss that eATP induces P2X7-dependent calcium signaling within cells, leading to an increase in proapoptotic signaling. Lung cancer cells with downregulated P2X7 are less sensitive to this mechanism. In line with this, Douguet and coworkers showed that the activation of P2X7 can inhibit the growth of NSCLCs in mice receiving an anti-PD-1 therapy and therefore renders the tumor cells sensitive to this therapy [57]. Consequently, we would have expected that high levels of P2X7 should lead to a prolonged PFS in our study population due to an increase in anti-tumor immune mechanisms. However, P2X7 function switches depending on the eATP concentration and the duration of signal activity. Upon activation by eATP, P2X7 works as a nonselective cation channel. Of note, prolonged activation leads to a formation of a macropore, which induces cell death [58]. The work by Benzaquen and colleagues could prove that cancer cells from ADCs express a nonfunctional P2X7-isoform, which is not capable of forming this characteristic macropore, resulting in persistence of malignant cells. At the same time, immune cells in the TME of ADCs retain a functionally active isoform of P2X7 [59]. These findings make the study of P2X7 in the NSCLC context even more complex since (i) P2X7 function is thus strongly influenced by eATP concentration/duration of activation and (ii) simple immunostaining cannot distinguish between the two P2X7 isoforms. Boldrini and co-workers could further establish that P2X7 expression is probably regulated by miRNAs within NSCLCs. They showed that high P2X7 expression is associated with a better PFS and OS and that miRNA-mediated downregulation of P2X7, especially in KRAS mutated tumors, impairs the prognosis of patients [60]. Sainz and colleagues recently summarized the different role of P2X7 in tumor versus immune cells and the sometimes-contradicting study results concerning its prognostic role in various cancers. There is, however, growing agreement in the community that P2X7 and related signaling pathways are also core mediators of tumor immune escape in the TME [61]. This underscores the importance of further studying P2X7 and its role in NSCLC and to consider the above-discussed details concerning its expression and function.

Despite the interesting findings of our study, there are several limitations of the presented approach. First, the study was designed as a monocentric, retrospective study and is therefore susceptible to bias. Second, only the abundance of the ectonucleotidases and purinergic receptors were examined, but not their biochemical activity or isoforms. Third, the analysis of the tumor surrounding stroma did not include a specific distinction of the TME cells into the various classes of immune cells and tissue-resident fibroblasts. Fourth, we did not analyze the samples for marker footprints that combine all the investigated proteins, i.e., we cannot say if there is a certain combination of ectonucleotidase and purine receptor expression with an exceptionally beneficial or adverse outcome.

In summary, our study found associations between the expressions of proteins from the purinergic system and patient outcomes in NSCLCs. Surprisingly, the ectonucleotidases CD39 and CD73 as well as the purinergic receptor P2X4 rather than P2X7 exhibited significant correlations with clinical outcome parameters. Further experimental studies are needed to clarify the causal relationships between these expression patterns and biological properties of the tumors. Similarly, the future role of targeted therapies addressing purinergic signaling in NSCLCs must be established in prospective interventional studies.

## 5. Conclusions

This study highlights the critical impact of purinergic signaling on NSCLC prognosis. High CD39, low CD73, and high P2X4 expression levels in tumor cells as well as high CD39, high CD73, and low P2X4 levels in the surrounding stroma are associated with a favorable prognosis. We hypothesize that the differential expression of the components of the purinergic systems finetunes the antineoplastic immune response in the TME. These findings support the therapeutic potential of targeting ectonucleotidases and purinergic receptors to artificially enhance anti-tumor immunological mechanisms to improve NSCLC outcomes.

## Figures and Tables

**Figure 1 cancers-17-01142-f001:**
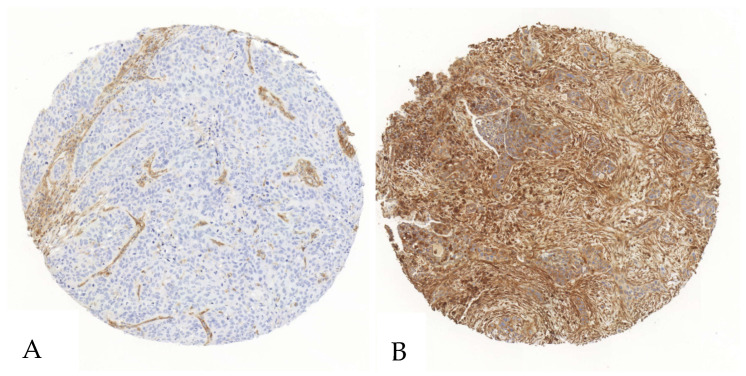
Exemplary illustration of CD39 expression in tumor cells and the surrounding stroma in a SCC tissue microarray core. (**A**). The tumor cells show no relevant positivity in a staining with anti-CD39-antibodies, whereas the stromal cells show intermediate positivity. (**B**). Both the stroma and the tumor cells show an intense staining when immunohistochemically stained with an anti-CD39-antibody.

**Figure 2 cancers-17-01142-f002:**
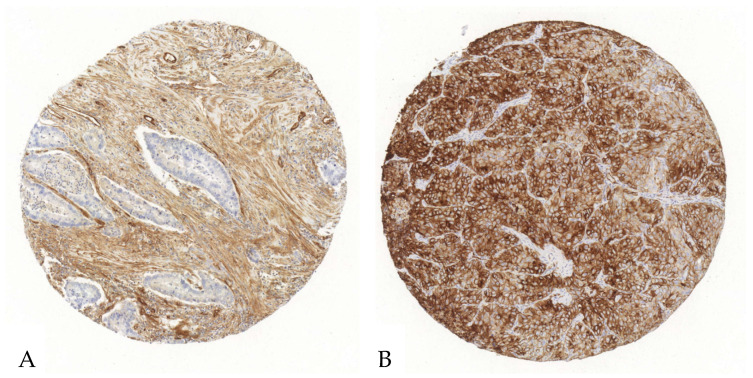
Exemplary illustration of CD73 expression in tumor cells and the surrounding stroma in an ADC tissue microarray core. (**A**). Upon immunohistochemical staining with an anti-CD73-antibody, the tumor cells reveal no relevant positivity, whereas the stroma shows a strong antibody expression. (**B**). Both the stroma and the tumor cells show an intense CD73-antibody expression.

**Figure 3 cancers-17-01142-f003:**
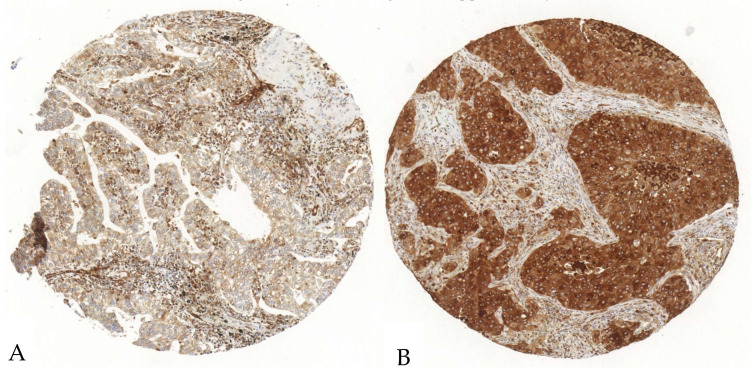
Exemplary illustration of P2X7 expression in tumor cells and the surrounding stroma in an ADC tissue microarray core. (**A**). The tumor cells show a scattered, weak to intermediate positivity, whereas the stroma shows a diffuse strong antibody expression. (**B**). The tumor cells show an intense P2X7-antibody expression; the stroma shows a diffuse intermediate expression.

**Figure 4 cancers-17-01142-f004:**
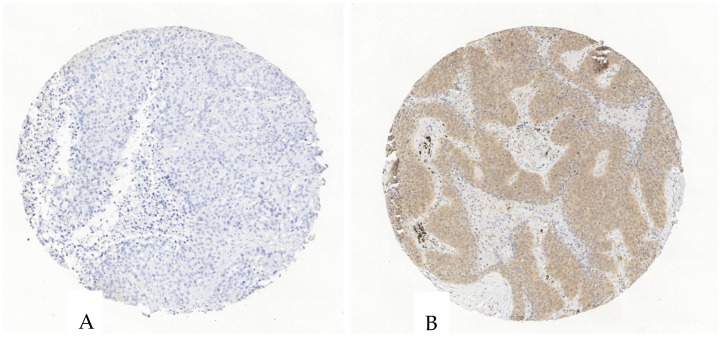
Exemplary illustration of P2X4 expression in tumor cells and the surrounding stroma in an ADC tissue microarray core. (**A**). The tumor and stroma cells are negative for P2X4. (**B**). The tumor cells show an intermediate P2X4-antibody expression, the stroma shows a scattered weak to intermediate expression.

**Figure 5 cancers-17-01142-f005:**
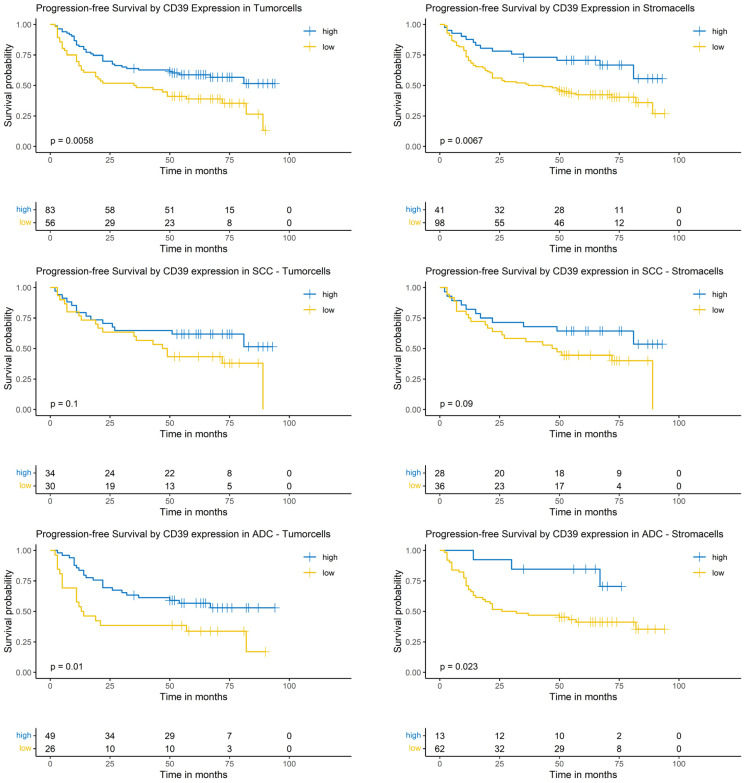
Kaplan–Meier-diagrams illustrating the correlation between CD39 abundance and progression-free survival (PFS) in tumor and surrounding stroma cells. **Upper** section: The diagrams depict the correlation between CD39 abundance in tumor cells (**left** panel) and stroma cells (**right** panel) and the PFS for the complete NSCLC cohort. A high CD39 expression (blue lines) in both histological compartments is significantly correlated with a prolonged PFS (*p* = 0.0058 for tumor cells and *p* = 0.0067 for stroma cells). **Middle** section: The diagrams show the correlation between CD39 abundance in tumor cells (**left** panel) and stroma cells (**right** panel) and the PFS for the squamous cell carcinoma (SCC) subpopulation. No statistically significant association between CD39 expression and PFS could be established in both histological compartments. **Lower** section: The diagrams depict the correlation between CD39 abundance in tumor cells (**left** panel) and stroma cells (**right** panel) and the PFS for the adenocarcinoma (ADC) subpopulation. A high CD39 expression (blue lines) in both histological compartments is significantly correlated with a prolonged PFS (*p* = 0.01 for tumor cells and *p* = 0.023 for stroma cells). The survival distributions were compared using the Logrank-test and results were considered statistically significant if *p* < 0.05.

**Figure 6 cancers-17-01142-f006:**
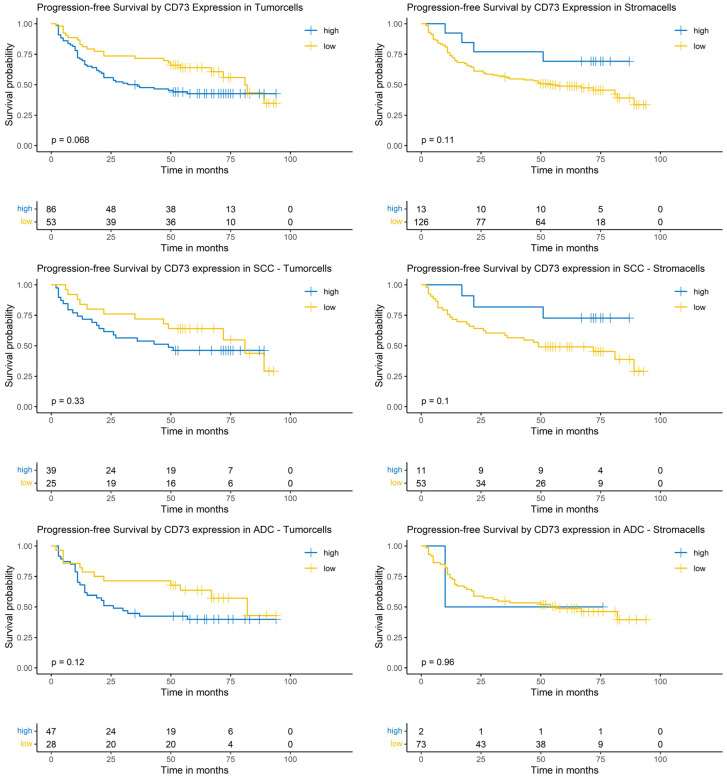
Kaplan–Meier-diagrams illustrating the correlation between CD73 abundance and progression-free survival (PFS) in tumor and surrounding stroma cells. **Upper** section: The diagrams depict the correlation between CD73 abundance in tumor cells (**left** panel) and stroma cells (**right** panel) and the PFS for the complete NSCLC cohort. No statistically significant association between CD73 expression and PFS could be established in both histological compartments. **Middle** section: The diagrams show the correlation between CD73 abundance in tumor cells (**left** panel) and stroma cells (**right** panel) and the PFS for the squamous cell carcinoma (SCC) subpopulation. No statistically significant association between CD73 expression and PFS could be established in both histological compartments. **Lower** section: The diagrams depict the correlation between CD73 abundance in tumor cells (**left** panel) and stroma cells (**right** panel) and the PFS for the adenocarcinoma (ADC) subpopulation. No statistically significant association between CD73 expression and PFS could be established in both histological compartments. The survival distributions were compared using the Logrank-test and the results were considered statistically significant if *p* < 0.05.

**Figure 7 cancers-17-01142-f007:**
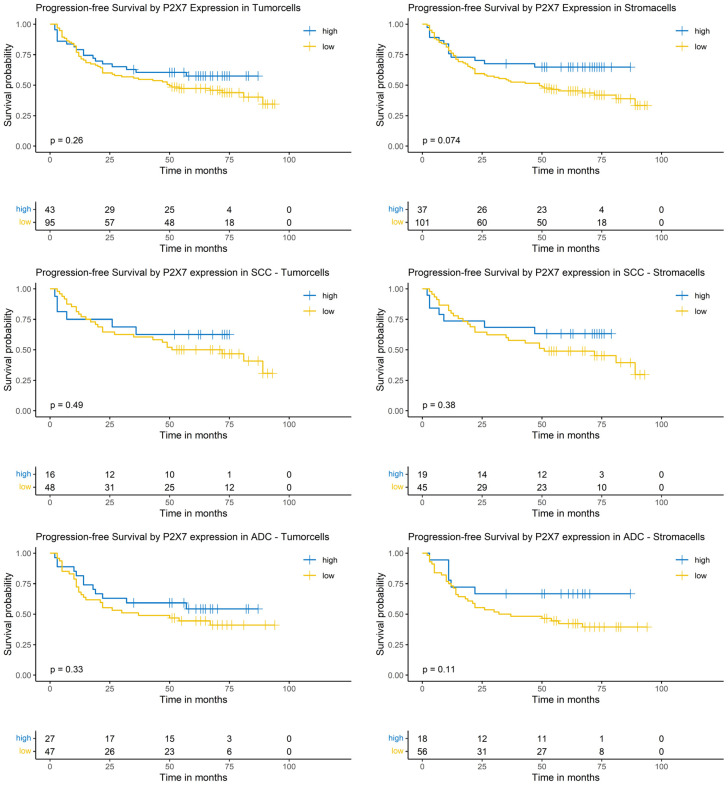
Kaplan–Meier-diagrams illustrating the correlation between P2X7 abundance and progression-free survival (PFS) in tumor and surrounding stroma cells. **Upper** section: The diagrams depict the correlation between P2X7 abundance in tumor cells (**left** panel) and stroma cells (**right** panel) and the PFS for the complete NSCLC cohort. No statistically significant association between P2X7 expression and PFS could be established in both histological compartments. **Middle** section: The diagrams show the correlation between P2X7 abundance in tumor cells (**left** panel) and stroma cells (**right** panel) and the PFS for the squamous cell carcinoma (SCC) subpopulation. No statistically significant association between P2X7 expression and PFS could be established in both histological compartments. **Lower** section: The diagrams depict the correlation between P2X7 abundance in tumor cells (**left** panel) and stroma cells (**right** panel) and the PFS for the adenocarcinoma (ADC) subpopulation. No statistically significant association between P2X7 expression and PFS could be established in both histological compartments. The survival distributions were compared using the Logrank-test and results were considered statistically significant if *p* < 0.05.

**Figure 8 cancers-17-01142-f008:**
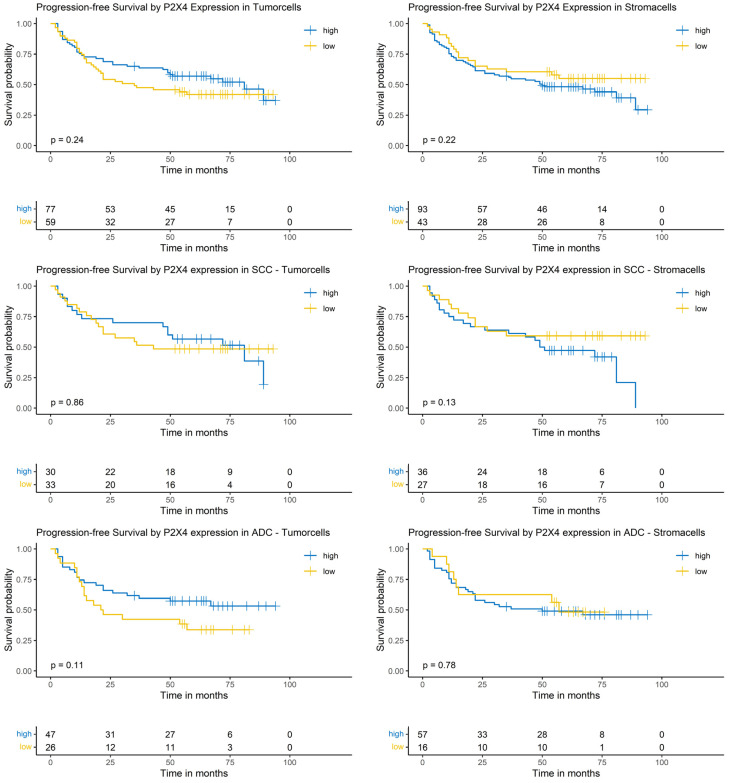
Kaplan–Meier-diagrams illustrating the correlation between P2X4 abundance and progression-free survival (PFS) in tumor and surrounding stroma cells. **Upper** section: The diagrams depict the correlation between P2X4 abundance in tumor cells (**left** panel) and stroma cells (**right** panel) and the PFS for the complete NSCLC cohort. No statistically significant association between P2X4 expression and PFS could be established in both histological compartments. **Middle** section: The diagrams show the correlation between P2X4 abundance in tumor cells (**left** panel) and stroma cells (**right** panel) and the PFS for the squamous cell carcinoma (SCC) subpopulation. No statistically significant association between P2X4 expression and PFS could be established in both histological compartments. **Lower** section: The diagrams depict the correlation between P2X4 abundance in tumor cells (**left** panel) and stroma cells (**right** panel) and the PFS for the adenocarcinoma (ADC) subpopulation. No statistically significant association between P2X4 expression and PFS could be established in both histological compartments. The survival distributions were compared using the Logrank-test and results were considered statistically significant if *p* < 0.05.

**Table 1 cancers-17-01142-t001:** Logrank-test of clinicopathological data and progression-free survival.

Logrank-Test	
Clinicopathological Parameter	*p*-Value
Age	0.086
Blood Vessel Invasion	0.0012 *
Histological Subtype	0.64
Lymph Vessel Invasion	<0.0001 *
Neoadjuvant Therapy	0.16
Pack Years	0.57
Perineural Sheath Invasion	0.24
pN-Classification	0.00068 *
pT-Classification	0.0052 *
Residual Disease	<0.0001 *
Sex	0.0036 *
Tumor Grade	0.81

Asterisks (*) indicate statistically significant results.

**Table 2 cancers-17-01142-t002:** Shapiro–Wilk-Test for analyzing the distribution of protein abundance in tumor and stroma cells. All protein expression patterns except for CD39 in tumor associated stromal cells showed a skewed distribution. An automated cutoff-finder (cutpointr package in “R”) was applied to define a cutoff for group dichotomization based on H-Scores.

H-Score	*p*-Value ^1^	Cutoff ^2^
CD39-Tumor	<0.01	4.2
CD39-Stroma	0.24	151.8
CD73-Tumor	<0.01	6.8
CD73-Stroma	<0.01	105.8
P2X7-Tumor	<0.01	205.7
P2X7-Stroma	<0.01	102.6
P2X4-Tumor	<0.01	12.49
P2X4-Stroma	<0.01	2.93

^1^ Shapiro-Wilk-Test, ^2^ Cutoff for dichotomization according to the H-Score.

## Data Availability

Dataset available on request from the authors.

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
