# Peer review of "The Ectonucleotidases CD39 and CD73 and the Purinergic Receptor P2X4 Serve as Prognostic Markers in Non-Small Cell Lung Cancer"

_cancers, 2025, doi:10.3390/cancers17071142_

Round 1
Reviewer 1 Report
Comments and Suggestions for Authors
Purinergic Receptors as Prognostic Markers in Lung Cancer
Konrad Kurowski1,2,3*, Sophie Nicole Prozmann4*, Antonio Eduardo Cabrita Figueiredo1, Jannis Heyer1,2,3, Felix 3 Kind5, Karl-Moritz Schröder1, Bernward Passlick4, Martin Werner1, 2, 3, Peter Bronsert1, 2, 3#, Severin Schmid4#
Major comments
The title does not translate the main results obtained as no clear correlation between P2X7 and P2X4 receptors expression and PFS was found. CD39 and CD73 are not purinergic receptors.
The authors claim that (and quoting) “The results show that high levels of CD39 and low levels of CD73 may help improve the body's immune response against the tumor, leading to better survival outcomes for patients”, however they do not provide such evidence.
In this study the authors present current data showing a correlation between higher tumor and stromal CD39 expression and lower tumor CD73 expression with prolonged PFS in patients with NSCLC.
How these observations correlate with the mechanism behind the expression/activity of these ectonucleotidases and their influence on ATP/ADP and AMP/ADO levels, activation of purinergic receptors and influence on cancer progression is not clearly discussed, explained or correlated. My suggestion is that these points should be better discussed in the light of current knowledge about the influence of these enzymes on cancer cell metabolism, cancer hallmarks and immune cell response.
Furthermore, the proposal of CD39 and CD73 as potential markers of NSCLC progression is not new and has been made by other authors with some conflicting results with the current study, which the authors should discuss (10.1136/jitc-2023-006770; 41419-023-06336-4; doi.org/10.3389/fimmu.2024.1427380).
In the current study the authors found no correlation between P2X7 and PFS. As in the case of the CD39 and CD73 markers, a figure with the distribution of P2X7 and PX4 receptors in tumor and stroma cells should also be included in the results section. The discussion of this topic should be improved and confronted with previous studies showing a correlation between PX7 receptor expression and PFS in patients with NSCLC adenocarcinoma (doi.10.3389/fimmu.2023.1287310; doi.10.3389/fphar.2020.00793; doi: .3892/ol.2014.2620).
The authors do not seem to provide a clear-cut result and discussion on the correlation between P2X4 receptors expression and PFS (section 3.13 of results and lines 417-422 of discussion). The reason behind the choice of studying the correlation between PFS and P2X4 receptors has been poorly supported and discussed.
Minor comments
Captions should be placed below the images. Figure captions should be more explanatory to effectively convey the information presented in the visual elements.
Reviewer 2 Report
Comments and Suggestions for Authors
Konrad Kurowski and collaborators, in the presented manuscript, investigated how the expression of specific receptors and enzymes involved in purinergic signaling affects patient outcomes. The rationale for the analysis is well justified, the results are quite interesting, and only some minor issues need to be addressed before the manuscript can be considered for publication.
The R package "cutpointr" has several method functions for cutpoint estimation; therefore, please provide more details on how the cutpoints were obtained.
Some of the presented data suggest that the assumption of proportional hazards may not have been met. Did the authors evaluate whether all their data satisfied this assumption? If so, please include the appropriate information in the Materials and Methods section. If not, please perform the analysis and adjust the test type for any data that did not meet the assumption.
The manuscript does not include information about p-value adjustments, which are necessary to maintain a family-wide type I error rate without inflation due to multiple comparisons (PMID: 30124010; PMID: 12471943). Please make the necessary corrections where applicable, particularly in the Cox analysis.
Reviewer 3 Report
Comments and Suggestions for Authors
There are many points must be improved before accept
1- In the abstract, the methods must be rewritten and numerical results need
2- The introduction is poor and must be improved
3- Add section of abberevation
4- All figures lignad must be below the figure
5-Histopathologic Parameters, formulate the results in table
6- Compare between the Expression in tumor cells and surrounding stroma for all Purinergic Receptors in table
Round 2
Reviewer 1 Report
Comments and Suggestions for Authors
The authors provided a significantly improved version of the manuscript and the main questions have been considered in the revised version of the manuscript.
Reviewer 3 Report
Comments and Suggestions for Authors
Accept in present form